# Association of hypertensive disorder of pregnancy with necrotizing enterocolitis in very preterm infants: A retrospective cohort study

Wenqian Chen[1‡], Jie Yang[2‡], Siyuan Jiang[3], Xiaoping Lei[4], Ligang Zhou[5], Jianguo Zhou[3], Liyuan Hu[3], Xinyue Gu[3], Cao Yun[3], Lizhong Du[6,7], Wenhao Zhou[3], Shoo Lee[8], Changyi Yang[1‡*], Yu Hu[9‡*], on behalf of Chinese Neonatal Network[¶]

1 Department of Neonatology, Fujian Maternity and Child Health Hospital, College of Clinical Medicine for Obstetrics & Gynecology and Pediatrics, Fujian Medical University, Fuzhou, Fujian, China, 2 NHC Key Laboratory of Neonatal Diseases, Children's Hospital of Fudan University, Shanghai, China, 3 Department of Neonatology, Children's Hospital of Fudan University, Shanghai, China, 4 Division of Neonatology, Department of Pediatrics, The Affiliated Hospital of Southwest Medical University, Luzhou, Sichuan, China, 5 Department of Pediatrics, Women and Children's Hospital of Chongqing Medical University, Chongqing, China, 6 Neonatal Intensive Care Unit, Children's Hospital, Zhejiang University School of Medicine, Hangzhou, China, 7 National Clinical Research Center for Child Health, National Children's Regional Medical Center, Hangzhou, China, 8 Maternal-Infant Care Research Centre and Department of Pediatrics, Mount Sinai Hospital, Toronto, Ontario, Canada, 9 Department of Pediatrics, Shengjing Hospital of China Medical University, Shenyang, Liaoning, China

‡ WC and JY contributed equally to this work as first author. CY and YH contributed equally to this work as correspondent author.
¶ Membership of the Chinese Neonatal Network is provided in the Acknowledgments.
* yangchangyi@fjmu.edu.cn (CY); huyu7678@163.com (YH)

**Data Availability Statement:** Due to the restrictions outlined in the "CHINESE NEONATAL NETWORK POLICIES AND PROCEDURES", specifically part 3 which states that "CHNN

## Abstract

Hypertensive disorders of pregnancy (HDP) may affect fetal development and result in preterm delivery. Necrotizing enterocolitis (NEC) is a severe gastrointestinal emergency in very preterm infants (VPIs, gestational age less than 32 weeks). The relationship between maternal HDP and NEC is controversial. Objective To investigate the association between maternal HDP and NEC in VPIs.This was a multicenter retrospective cohort study based on the data from the Chinese Neonatal Network (CHNN) which were collected between January 1, 2019 and December 31, 2021. Preterm infants born between 24+0 and 31+6 weeks of gestation were divided into HDP and no-HDP groups according to the 2015 Chinese guidelines for HDP. The primary outcome was the incidence of Bell's stage II or higher NEC. Secondary outcomes included mortality and spontaneous intestinal perforation (SIP). Of 27,660 women were included in the study analysis, 5405 (19.5%) were HDP and 22256 (80.5%) were no-HDP. NEC occurred in 5.2% (283/5,404) among HDP mothers and 5.3% (1,191/22,256) among no-HDP mothers. No significant association was observed between HDP and Bell's stage II or higher NEC (aOR 0.87, 95% CI [0.72, 1.05]). However, even after adjustment, maternal HDP appeared to be protective for NEC requiring surgical intervention (aOR 0.60, 95% CI [0.43, 0.83]). There was no significant correlation between maternal HDP and neonatal mortality and SIP. Maternal HDP was not significantly associated with

coordinating center will not release individual patient data to any investigator and all analyses will be conducted at CHNN Coordinating Center," the authors do not have the right to publicly release the data. Usage of the data requires approval from the Ethics Committee and the CHNN Executive Committee. For more information, please contact the CHNN Data Committee (chnndata@163.com).

**Funding:** The author(s) received no specific funding for this work.

**Competing interests:** The authors have declared that no competing interests exist.

the incidence of Bell's stage II or higher NEC. However, it was associated with the lower rate of NEC requiring surgical intervention.

## Introduction

Necrotizing enterocolitis (NEC) is a prevalent gastrointestinal crisis in preterm infants, with a global incidence rate of about 7% [1]. The incidence rate of NEC in preterm infants born less than 32 weeks of gestation in China is about 5.5% [2]. NEC is primarily managed through medical treatment, with approximately 30% to 50% of severe NEC cases that do not respond to conservative medical management requiring surgical intervention [3]. Despite aggressive medical and surgical interventions, the mortality rate of NEC remains as high as 25% [4]. Survivors are prone to long-term sequelae, including short bowel syndrome and neurodevelopmental delays, significantly impacting the prognosis of premature infants [5]. NEC is a multifactorial disease associated with factors such as preterm birth, low birth weight, and formula feeding [6]. However, there is limited research on the relationship between maternal diseases and NEC.

Hypertensive disorder of pregnancy (HDP) is indeed one of the most prevalent complications that can occur during pregnancy [7]. Approximately 10% of pregnant women experience HDP [8], and its incidence has been on the rise over the past few decades [9]. The incidence of HDP in China is higher than that in other parts of the world [10], with a rate of up to 18.8% in premature infants with a gestational age between $24^{+0}$ and $31^{+6}$ weeks whose mothers have HDP [2]. HDP causing placental and umbilical blood flow abnormalities may lead to feeding issues in preterm infants [11, 12]. However, the relationship between HDP and NEC has been controversial. Most of the conclusions about the relationship between HDP and NEC are based on studies of overall adverse outcomes in newborns, in which NEC is only observed as a secondary outcome after HDP exposure and the influence of many factors that affect the occurrence of NEC (such as small for gestational age(SGA), gestational age(GA), and feeding methods) are not considered [13].

Hence, the objective of this study was to assess the correlation between HDP and NEC in very preterm infants (VPIs, born less than 32 weeks of gestation), utilizing data from the Chinese Neonatal Network(CHNN), which represents the most extensive cohort of preterm infants in China.

## Materials and methods

### Study design and setting and data collection

This was a retrospective cohort study using data prospectively collected in the CHNN database between January 1, 2019 and December 31, 2021. CHNN maintains a standardized national clinical database of very preterm or very low birth weight infants (GA < 32 weeks or birth weight < 1500 g) admitted to tertiary neonatal intensive care units (NICUs). The primary purpose is to monitor changes in outcomes and care practices and explore strategies for enhancing neonatal care [14, 15]. The data were accessed for research purposes beginning January 1, 2019. In 2019, 2020, and 2021, there were 57, 70, and 79 NICUs participated in CHNN and collected whole-year data of all admitted very preterm or very low birth weight infants. These NICUs were all tertiary referral facilities and were carefully chosen to represent neonatal care in various regions of the country.

Trained abstractors at each site recorded information directly from patient charts into a customized database. Patient information is kept confidential, ensuring that researchers do not have access to any information that could identify individual participants during or after data collection. Following, the data were sent electronically to the CHNN coordinating center at Children's Hospital of Fudan University. The database had error-checking mechanisms, standard protocols, and a definitions manual. At the coordinating center, data quality and integrity were rigorously assessed through audits to ensure accuracy and reliability. These measures aimed to maintain high data quality and enhance the study's findings' credibility [16].

This study received approval from the ethics review board of Children's Hospital of Fudan University (2018–296), and this approval was recognized by all participating hospitals. Waiver of consent was granted at all sites as the study utilized deidentified patient data. Furthermore, the study adhered to the Strengthening the Reporting of Observational Studies in Epidemiology (STROBE) reporting guideline for cohort studies, ensuring transparency and comprehensive reporting of the study's findings.

## Participants

The study population included newborns who were born between $24^{+0}$ and $31^{+6}$ gestational weeks and were admitted to the CHNN participating hospitals. Stillbirths, neonatal deaths in the delivery room, and newborns transferred to non-participating hospitals within 24 hours of birth were not included in the CHNN database. Congenital malformation, patients with missing data on maternal HDP and NEC were excluded.

## Exposure

The diagnosis criteria for HDP in this study followed the 2015 Chinese guidelines for hypertensive disorders of pregnancy. HDP was defined as having a systolic blood pressure $\geq 140$ mmHg and/or a diastolic blood pressure $\geq 90$ mmHg, including both chronic hypertension and pregnancy-induced hypertension [17]. Preeclampsia is a more severe form of hypertensive disorder characterized by high blood pressure, proteinuria, and potential organ dysfunction, while eclampsia refers to the occurrence of seizures in a woman with preeclampsia [18]. Preeclampsia/eclampsia data were collected for HDP mothers based on their obstetric records, determining the presence or absence of preeclampsia or eclampsia.

## Outcomes

The primary outcome was NEC stage II or III, defined according to Bell's criteria. Surgical necrotizing enterocolitis (SNEC) was defined as NEC that required surgical treatment. Pneumoperitoneum due to intestinal perforation is the absolute indication for NEC surgical treatment. Relative indications for NEC surgical treatment include: 1. Stage IIIa of Bell stage for diagnosis of NEC is ineffective after 48 hours of conservative treatment or stage IIIb, accompanied by oliguria, hypotension, and metabolic acidosis that is difficult to correct. 2. Abdominal X-ray examination revealed intestinal rigidity and fixation with portal vein gas. 3. If intestinal perforation is highly suspected, but abdominal X-ray examination does not find pneumoperitoneum, if the abdominal drainage is yellowish brown turbidous liquid.

Secondary outcomes included mortality and spontaneous intestinal perforation (SIP). Mortality refers to mortality during hospitalization. SIP refers to the occurrence of intestinal perforation without any apparent external cause or mechanical force.

## Covariates

Gestational age (GA) was ascertained through a hierarchical approach based on the best obstetric estimate, which considered prenatal ultrasound, menstrual history, obstetric examination, or a combination of all three methods. In cases where the obstetric estimate was unavailable or differed from the postnatal estimate of gestation by more than 2 weeks, the GA was estimated using the Ballard Score [19]. Small for gestational age (SGA) was defined as birth weight and length below the 10th percentile for GA [20]. Antenatal steroid was defined as one or more dose administered before delivery, regardless of the type of steroid. Breastfeeding is defined as exclusive breastfeeding during hospitalization, no formula feeding.

## Statistical analysis

The statistical description of continuous variables was presented using means with standard deviations (SD) or medians with interquartile ranges (IQR). Categorical variables were described using percentages. Comparisons between groups were performed using the student's t-test and Mann-Whitney U test for continuous data, and the Chi-square test or Fisher's exact test for dichotomous data. Multivariable logistic regression models were used to examine the association between HDP and different outcomes with adjusted for key confounders and odds ratio (OR). Sensitivity analysis explored different models. Model 1 adjusted for key confounders: GA, Birth weight (BW), Antenatal steroids, magnesium sulfate, Chorioamnionitis, Breast feeding, and Patent ductus arteriosus (PDA). Model 2 expands on Model 1, adding ROM ≥24, Cesarean delivery, Twin/Multiple births, maternal age, while excluding Breast feeding and PDA. Model 3 builds on Model 2, including Male gender, PDA, Apgar scores ≤5 at 7 minutes, and Breast feeding. Subgroup analysis were conducted between different GA, BW, SGA, number of fetuses, and place of delivery to explore the correlation between HDP and NEC in subgroup population. Multiple imputation was not applied since the missing data were <10%. Risk estimates were presented as 95% confidence interval (CI). Two-sided p-values less than 0.05 were considered to be statistically significant. Statistical analyses were conducted using R software package (version R 4.2.2).

## Results

A total of 28,264 VPIs were studied. As there were no cases of NEC missing, after excluding 255 cases with multiple anomalies and 349 cases with missing maternal HDP data, 27,660 infants were analyzed. Among them, 5,404 (19.5%) whose mother had HDP, while 22,256 (80.5%) without HDP. Among infants born to HDP mothers, 283 (5.2%) developed NEC, compared to 1,191 (5.3%) in newborns without HDP mothers (Fig 1).

Compared to no-HDP mothers, pregnant women with HDP were older in age, had a higher rate of receiving corticosteroids and magnesium sulfate treatment during the antenatal period, and had a lower rate of twin/multiple pregnancies, less prolonged rupture of membranes ≥24 hours, and chorioamnionitis. Newborns of mothers with HDP were more likely to be delivered by cesarean section, had slightly older gestational ages, lower birth weights, a significantly higher incidence of SGA, and a lower proportion of male infants. There was a similar proportion of 5-minute Apgar scores ≤7 and occurrence of postnatal patent ductus arteriosus between the two groups. The majority of infants in the HDP group were born in the obstetrics hospital, and the exclusive breastfeeding rate was slightly lower (Table 1).

Newborns born to mothers with HDP demonstrated a reduced occurrence of NEC stage III (OR 0.74[0.59,0.92]) and SNEC (OR 0.65[0.49,0.85]), as well as a lower mortality rate, compared to newborns without HDP in their mothers. There is no significant difference between the two groups in the incidence of NEC stage II and SIP. After adjusting for GA, BW,

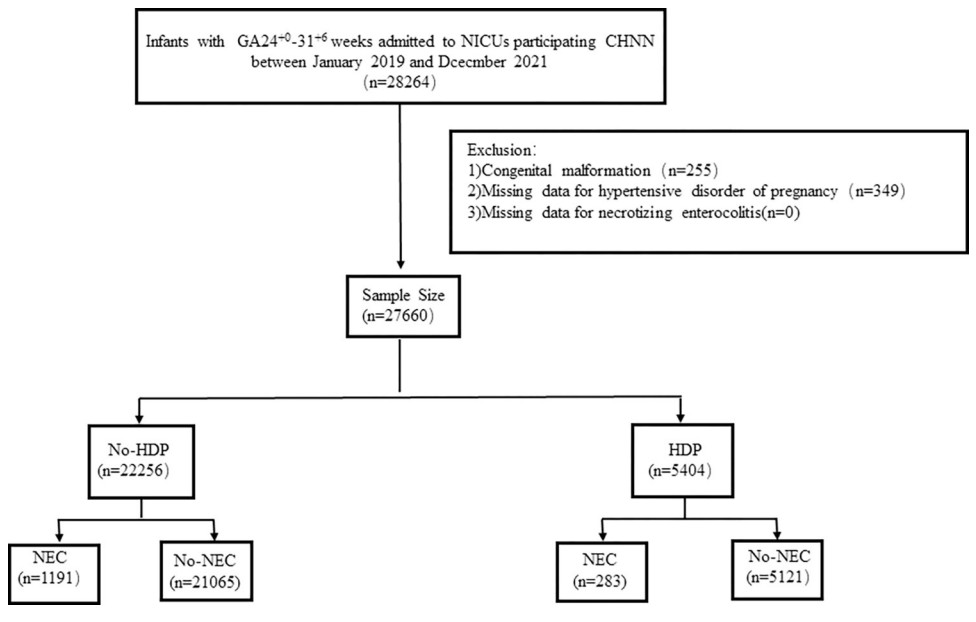

**Fig 1. Flow chart for patient selection.**

**Table 1. Baseline characteristics of participants.**

| Variables | HDP (N = 5404) | No-HDP (N = 22256) | P value |
|---|---|---|---|
| **Maternal characteristics, n/N (%)** | | | |
| Maternal age (years), mean (SD) | 32.2 (5.0) | 30.6 (4.9) | <0.01 |
| Twin/Multiple birth | 780/ 5404 (14.3) | 6496/22256 (29.0) | <0.01 |
| Antenatal steroids | 4061/ 5032(80.7) | 15622/20406(76.6) | <0.01 |
| Antenatal MgSO4 | 3419/4742 (72.1) | 9360/19234(48.7) | <0.01 |
| PROM ≥24hr | 239/5138(4.7) | 5772/20860 (27.7) | <0.01 |
| Chorioamnionitis | 235/3984(5.9) | 2609/15475(16.9) | <0.01 |
| Cesarean | 4924/ 5391(91.3) | 11007/ 22209(49.5) | <0.01 |
| Inborn | 3832/5404(71.0) | 14516/22256(65.0) | <0.01 |
| **Infant characteristics, n/N (%)** | | | |
| Gestational age, (weeks), median (IQR) | 30.1(29.0–31.1) | 29.7(28.1–30.9) | <0.01 |
| <28 | 466/5404(8.6) | 4485/22256(20.2) | <0.01 |
| ≥28 | 4938/5404(91.4) | 17771/22256(79.8) | <0.01 |
| Birth weight (g), mean (SD) | 1188.4(282.7) | 1337.9(323.5) | <0.01 |
| <1000 | 1406/5404(26.0) | 3348/22256(15.0) | <0.01 |
| 1000–1500 | 3261/5404(60.0) | 11780/22256(53.0) | <0.01 |
| ≥1500 | 737/5404(14.0) | 7128/22256(32.0) | <0.01 |
| Male | 2786/5397(51.6) | 12873/22239(57.9) | <0.01 |
| SGA | 1353/5404(25.0) | 766/22256(3.4) | <0.01 |
| Apgar score at 5 min ≤7 | 757/5154 (14.8) | 3109/21052(15.5) | 0.90 |
| PDA | 2453/5271(46.5) | 10016/21559(46.5) | 0.90 |
| Breast feeding | 679/5404 (13.0) | 3412/22256 (15.0) | <0.01 |

PROM, premature rupture of membranes; MgSO4,magnesium sulfate; SGA, small for gestational age(defined as birth weight and length below the 10th percentile for gestational age);PDA, patent ductus arteriosus

**Table 2. Outcomes of preterm infants born among mothers with and without HDP.**

| Outcomes n (%) | HDP (N = 5404) | No-HDP (N = 22256) | P value | Unadjusted OR (95% CI) | Adjusted OR (95% CI) [a] | Adjusted OR (95% CI) [b] | Adjusted OR (95% CI) [c] |
|---|---|---|---|---|---|---|---|
| NEC ≥II | 283 (5.2) | 1191 (5.4) | 0.70 | 0.98(0.85,1.11) | 0.87(0.72,1.05) | 0.85(0.69,1.05) | 0.85 (0.69,1.05) |
| NEC-II | 174 (3.2) | 684 (3.1) | 0.60 | 1.05(0.88,1.24) | 0.91(0.72,1.14) | 0.90(0.70,1.15) | 0.91 (0.70,1.17) |
| NEC-III | 100 (1.9) | 473 (2.1) | 0.20 | 0.74(0.59,0.92) | 0.78(0.56,1.08) | 0.77(0.53,1.10) | 0.75(0.51,1.08) |
| SNEC | 93 (1.7) | 512(2.3) | <0.01 | 0.65(0.49,0.85) | 0.60(0.43,0.83) | 0.58(0.40,0.83) | 0.58(0.39,0.82) |
| Mortality | 554 (10.3) | 2514 (11.3) | 0.02 | 0.90(0.81,0.99) | 1.07(0.92,1.25) | 1.12(0.95,1.31) | 1.04(0.87,1.24) |
| SIP | 56 (1.0) | 220 (1.0) | 0.70 | 1.05(0.78,1.40) | 1.22(0.76,1.90) | 1.05(0.63,1.69) | 1.07(0.63,1.76) |

HDP, Hypertensive disorder of pregnancy; NEC, necrotizing enterocolitis; SNEC, surgery NEC; SIP, Spontaneous intestinal perforation

[a] model 1 adjusted for Gestational age, Birth weight, Antenatal steroids, MgSO4, Chorioamnionitis, Breast feeding,PDA.

[b] model 2 adjusted for Gestational age, Birth weight, Antenatal steroids, MgSO4, Chorioamnionitis, PROM≥24hr, Cesarean, Twin/Multiple birth, Maternal age.

[c] model 3 adjusted for Gestational age, Birth weight, Antenatal steroids, MgSO4, Chorioamnionitis, Maternal age, Cesarean, Twin/Multiple birth, PROM ≥24hr,Male, PDA,Apgar5≤7, Breast feeding.

Antenatal steroids, magnesium sulfate, Chorioamnionitis, Breast feeding and PDA, maternal HDP showed no significant correlation with the occurrence of NEC stage II and III (aOR 0.87 [0.72,1.05]), mortality, or SIP in premature infants. However, as the severity of NEC increases, the impact of HDP on NEC becomes more pronounced. Maternal HDP are linked to a decrease of approximately 40% in the likelihood of NEC requiring surgical intervention (aOR 0.60 [0.43,0.83]). After adding confounding factors such as Maternal age, Cesarean, Twin/Multiple birth, premature rupture of membranes more than 24hr, Male, and Apgar5≤7 to the multiple regression model, the results remained stable (Table 2).

Subgroup analysis based on GA, birth weight, SGA, fetal number, and delivery location revealed that the mother with HDP was not associated with NEC stages II or III between subgroup population (Fig 2, S1 Table).

Compared to premature infants without HDP, maternal preeclampsia/eclampsia reduces the risk of SNEC (aOR 0.56[0.38,0.81]). However, in HDP cases without preeclampsia/eclampsia, this effect is not observed (Table 3).

## Discussion

We evaluated the influence of maternal HDP on the occurrence of NEC in VPIs, based on data from CHNN, the largest multicenter neonatal cohort in China. Our study found no significant association between maternal HDP and the occurrence of stages II or III NEC in VPIs. However, we observed a gradual and significant increase in the impact of maternal HDP on NEC

**Table 3. The association of subgroup of HDP with outcomes.**

| Outcomes | pre- and/or eclampsia (1)N = 4044 | HDP without pre- and/or eclampsia (2)N = 1218 | No-HDP (3) N = 22398 | Adjusted OR/ (95%) [a] (1)vs (3) | Adjusted OR (95%) [a] (2)vs (3) |
|---|---|---|---|---|---|
| NEC ≥II,n/N (%) | 209/4044(5.2) | 68/1218(5.5) | 1197/22398(5.3) | 0.82(0.66,1.01) | 1.04(0.76,1.41) |
| NEC-II,n/N (%) | 130/4044(3.2) | 42/1218 (3.4) | 686/22398 (3.0) | 0.86(0.66,1.11) | 1.13(0.76,1.61) |
| NEC-III,n/N (%) | 70/4044(1.7) | 26/1218 (2.1) | 477/22398 (2.1) | 0.72(0.49,1.04) | 0.94(0.53,1.56) |
| SNEC,n/N (%) | 72/4044(1.8) | 17/1218 (1.4) | 516/22398 (2.3) | 0.56(0.38,0.81) | 0.70(0.38,1.19) |

HDP, Hypertensive disorder of pregnancy; NEC, necrotizing enterocolitis; SNEC, surgery NEC; a adjusted for Gestational age, Birth weight, Antenatal steroids, MgSO4, Chorioamnionitis, Breast feeding, PDA.

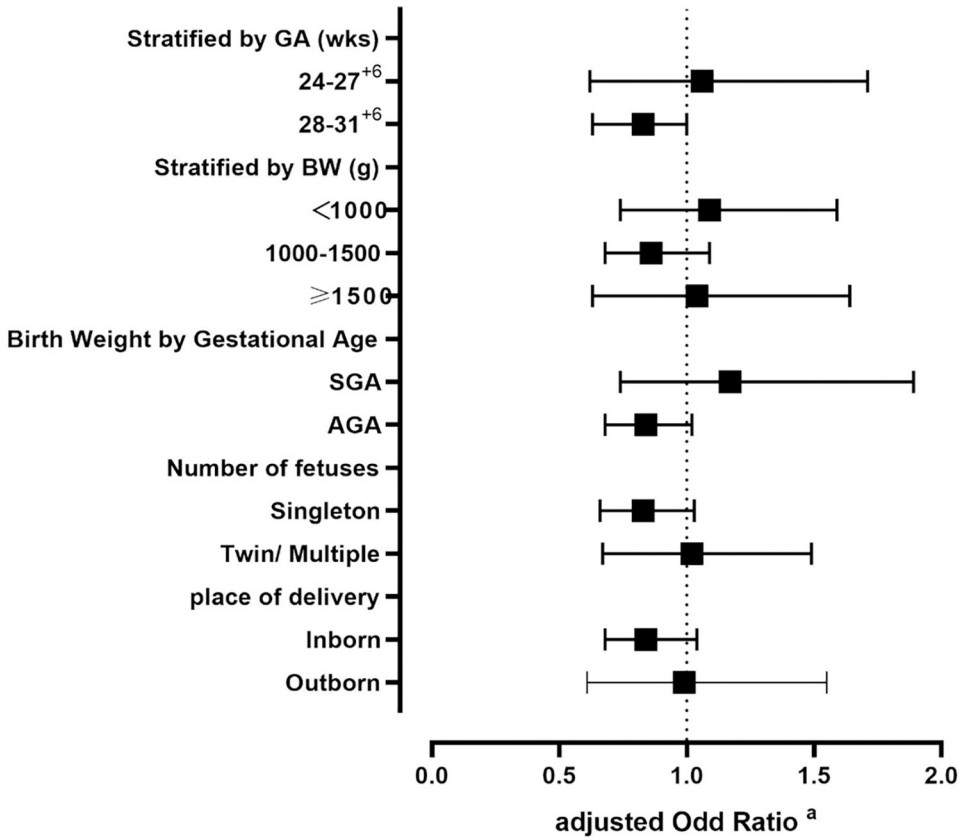

**Fig 2. Subgroup analysis for the association between HDP and NEC≥II, stratified by GA,BW, birth weight by gestational age, number of fetuses and place of delivery.** a adjusted for Birth weight, Antenatal steroids, MgSO4, Chorioamnionitis, Breast feeding, PDA.

severity. HDP reduced the risk of SNEC by approximately 40%, even after accounting for known risk factors such as GA, birth weight, antenatal magnesium sulfate administration, antenatal corticosteroid administration, chorioamnionitis, breastfeeding, and PDA. There was no significant correlation between maternal HDP and mortality or SIP.

Our research findings align with a German multicenter study, showing a significant reduction in SNEC in extremely low birth weight infants born before 32 weeks of gestation with maternal preeclampsia and HELLP syndrome [21]. Study comparing adverse outcomes of preterm births due to HDP and chorioamnionitis found lower risk of SNEC and intestinal perforation in HDP-related preterm births [22]. Regev et al. used the Israel Neonatal Network database to study the impact of maternal HDP on SGA preterm infants (gestational age 24 to 32 weeks). They found a reduced risk of NEC in newborns of mothers with preeclampsia, excluding other pregnancy complications like antepartum hemorrhage, chorioamnionitis, and prolonged rupture of membranes [23]. The reduced SNEC incidence in preterm infants of HDP mothers may be due to better antenatal attention and treatment. The HDP group received more corticosteroids and magnesium sulfate therapy, with lower rates of premature rupture of membranes and chorioamnionitis. Adequate monitoring and management during pregnancy are crucial for preventing adverse outcomes for both mothers and newborns [24, 25]. It has been found that antenatal corticosteroid use reduces the incidence and severity of NEC [6]. Antenatal magnesium sulfate use exhibited a time-dependent reduction in the incidence of intestinal diseases requiring surgery in preterm infants [26]. Premature rupture of

membranes and chorioamnionitis in mothers will lead to an increase in the colonization of Escherichia coli in preterm infants. This is associated with an increased incidence of NEC. In our study, there was no difference in stage III NEC after adjustment, but the risk of SNEC was reduced, which may be related to the relatively mild intestinal injury in the HDP group and the good response to medical treatment. The exact mechanisms by which HDP reduces SNEC risk are unclear. However, the ischemic preconditioning mechanism may play a role in this process. A study on the effects of ischemic preconditioning in a neonatal NEC model in pregnant mice found that ischemic treatment given to the pregnant mice 24 hours before delivery had a protective effect on the intestines of the hypoxia-reoxygenation neonatal rats. It reduced the severity of intestinal mucosal injury and maintained proliferative activity [27]. Preeclampsia/eclampsia is believed to be caused by invasive changes in the extravillous trophoblast cells and insufficient remodeling of the spiral arteries in the maternal placenta, leading to inadequate placental perfusion during development [28].

In our study, mothers with preeclampsia/eclampsia had a reduced SNEC risk in premature infants compared to those without HDP. However, among infants born to mothers with HDP but without preeclampsia/eclampsia, there was no significant difference in SNEC incidence. This suggests that preeclampsia/eclampsia in mothers with HDP may confer a protective effect against SNEC in premature infants, while HDP alone, without preeclampsia/eclampsia, might not provide the same benefit. This may be attributed to different etiologies of HDP and varying degrees of placental pathology. Jerzy Stanek [29] found that severe preeclampsia/eclampsia was mostly associated with placental pathology, whereas non-preeclamptic/eclamptic cases were predominantly related to maternal diseases rather than placental abnormalities. Placental pathology in cases of preeclampsia/eclampsia resulted in ischemic preconditioning in the infant intestines.

There are also varying reports on the association between HDP and NEC. Perger et al. suggested that preeclampsia is an independent risk factor for NEC [30]. However, some recent cohort studies on HDP and adverse neonatal outcomes have found no correlation between HDP and NEC [2]. These studies often treated NEC as a secondary outcome and did not account for various confounding factors, including breastfeeding, chorioamnionitis, SGA, corticosteroid administration, and magnesium sulfate therapy, potentially impacting the findings.

The relationship between HDP and neonatal mortality is still controversial. Rocha's cohort study on preterm births below 34 weeks of gestation found that HDP increased the risk of mortality [24]. In contrast, a multicenter study in Canada reported that mothers exposed to HDP had a lower mortality rate in SGA infants compared to other causes [31]. Our study similarly suggests that maternal HDP exposure may reduce the risk of mortality in the $24^{+0}$ to $31^{+6}$ weeks preterm population, although the difference was not statistically significant. This finding aligns with some previous studies [2]. The cause of SIP is currently unknown but has been linked to maternal chorioamnionitis [32]. Limited research exists on the correlation between HDP and SIP. In our study, we found no significant correlation between HDP and SIP, which contradicts the findings of Yılmaz et al. [33]. However, their study had a small sample size and did not include chorioamnionitis in the multivariable analysis.

Our study has limitations including: 1. Due to closer monitoring in HDP pregnancies and differences in surgical NEC indications among centers, there may be intervention bias. 2. Using existing data, we can't account for unmeasured confounding factors like smoking and alcohol. However, smoking and alcohol consumption among Chinese women are rare. The smoking rate among women aged 20–29 is less than 2% according to a national survey [34]. 3. Since our database did not collect relevant information on miscarriages and stillbirths, we cannot determine the rate of intrauterine fetal death in HDP pregnancies, which may introduce selection bias. Nevertheless, the CHNN database is a robust and standardized data source,

offering reliable and validated information. Its large population size and detailed demographic data allow effective adjustment for numerous confounding variables. This enhances the study's validity and generalizability.

## Supporting information

**S1 Table. Subgroup analysis for the association between HDP and NEC≥II, stratified by GA, BW, birth weight by gestational age, number of fetuses and place of delivery.** [a] adjusted for Birth weight, Antenatal steroids, MgSO4, Chorioamnionitis, Breast feeding, PDA. [b] adjusted for Gestational age, Antenatal steroids, MgSO4, Chorioamnionitis, Breast feeding, PDA. [c] adjusted for Antenatal steroids, MgSO4, Chorioamnionitis, Breast feeding, PDA. [d] adjusted for Birth weight, Gestational age, Antenatal steroids, MgSO4, Chorioamnionitis, Breast feeding, PDA.
(DOCX)

## Acknowledgments

Our gratitude extends to the data abstractors from the Chinese Neonatal Network for their valuable contributions. We also extend our thanks to all the staff at the Chinese Neonatal Network coordinating center for their unwavering organizational support. We thank Ruimiao Bai from Northwest Women's and Children's Hospital, Dan Dang from The First Hospital of Jilin University, Juan Du from Beijing Children's Hospital, Wei Shi from Zhejiang University School of Medicine, Min Yang from Obstetrics and Gynecology Hospital of Fudan University, Wenli Li from The Third Affiliated Hospital of Zhengzhou University, Yujie Han from Children's Hospital Affiliated to Shandong University, Aimin Qian from Children's Hospital of Nanjing Medical University, Dan Zhao from Maternal and Child Health Hospital of Guangxi Zhuang Autonomous Region, Pei Lu from Shanghai Children's Hospital, Mengya Sun from the Affiliated Hospital of Qingdao University, Ru Xue from Shanghai Children's Medical Center, Yuru Zhu from Gansu Provincial Maternity and Child Care Hospital, Ping Cheng from Henan Children's Hospital, Yuanyuan Chen, Mengmeng Ge and Shujuan Li from Children's Hospital of Fudan University, and Li Wang from Maternal and Children' Hospital of Chongqing Medical University for their guidance on this article.

## Author Contributions

**Conceptualization:** Wenqian Chen, Jie Yang, Changyi Yang, Yu Hu.

**Data curation:** Jie Yang, Siyuan Jiang, Xinyue Gu, Cao Yun, Lizhong Du, Wenhao Zhou, Yu Hu.

**Formal analysis:** Wenqian Chen, Jie Yang, Siyuan Jiang, Xinyue Gu, Wenhao Zhou, Yu Hu.

**Funding acquisition:** Xinyue Gu, Wenhao Zhou.

**Investigation:** Wenqian Chen, Jie Yang, Siyuan Jiang, Xiaoping Lei, Ligang Zhou, Jianguo Zhou, Liyuan Hu, Xinyue Gu, Changyi Yang, Yu Hu.

**Methodology:** Wenqian Chen, Jie Yang, Siyuan Jiang, Xiaoping Lei, Ligang Zhou, Jianguo Zhou, Liyuan Hu, Changyi Yang, Yu Hu.

**Project administration:** Cao Yun, Lizhong Du, Wenhao Zhou, Shoo Lee, Changyi Yang.

**Resources:** Wenqian Chen, Siyuan Jiang, Xinyue Gu, Cao Yun, Shoo Lee, Changyi Yang, Yu Hu.

**Software:** Wenqian Chen, Jie Yang, Xinyue Gu.

**Supervision:** Xinyue Gu, Cao Yun, Lizhong Du, Wenhao Zhou.

**Validation:** Wenqian Chen, Jie Yang.

**Visualization:** Wenqian Chen.

**Writing – original draft:** Wenqian Chen, Jie Yang, Changyi Yang, Yu Hu.

**Writing – review & editing:** Wenqian Chen, Jie Yang, Changyi Yang, Yu Hu.

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
