## [Decision Letter · Decision Letter 0]

11 Jul 2024

PONE-D-24-12553Association of hypertensive disorder of pregnancy with necrotizing enterocolitis in very preterm infants: A retrospective cohort studyPLOS ONE

Dear Dr. Hu,

Thank you for submitting your manuscript to PLOS ONE. After careful consideration, we feel that it has merit but does not fully meet PLOS ONE’s publication criteria as it currently stands. Therefore, we invite you to submit a revised version of the manuscript that addresses the points raised during the review process.

We look forward to receiving your revised manuscript.

Kind regards,

Renato Teixeira Souza

Academic Editor

PLOS ONE

Journal Requirements:

3. In the online submission form, you indicated that the datasets used in this article are not publicly available due to the inclusion of part of the data in another article that is currently under preparation. If you wish to access the datasets, please direct your requests to Yun Cao at yuncao@fudan.edu.cn. They will be able to assist you further regarding the data availability and access. 

Additional Editor Comments:

I would suggest considering it for publication in a forthcoming issue if the authors address the minor revisions recommended.

Reviewers' comments:

Reviewer's Responses to Questions

**Comments to the Author**

1. Is the manuscript technically sound, and do the data support the conclusions?

Reviewer #1: Yes

Reviewer #2: Yes

2. Has the statistical analysis been performed appropriately and rigorously? 

Reviewer #1: Yes

Reviewer #2: Yes

3. Have the authors made all data underlying the findings in their manuscript fully available?

Reviewer #1: Yes

Reviewer #2: No

4. Is the manuscript presented in an intelligible fashion and written in standard English?

Reviewer #1: Yes

Reviewer #2: Yes

5. Review Comments to the Author

Reviewer #1: In this manuscript the authors studied the relationship between maternal HDP and the babies NEC using database of CHNN registry.

The result suggests maternal HDP and NEC have not significantly affected each other, but the rate of NEC with surgical treatment is associated Maternal HDP.

Generally, the result of this manuscript is very interesting.

Please refer to a comment as follows, it is a minor point for revision.

L177-L178 ;

It would be better to add the description about the missing data of NEC, although it is described in Fig 1.

Reviewer #2: The manuscript approaches a very important theme in perinatology. Methodology is weel-designed, with a sufficient number of participants for a cohort study. Most of the data was well explained at dicussion section.

Nevertheless, there are some mild considerations to be analysed:

1) Please consider changing the word "conference" at line 172.

2) Your results show that, after ajustments, there was no significant correlation between maternal HDP and the occurence of NEC stages II or III. However, at Discussion section, it's said that this study revealed that VPIs born to mothers with HDP had a lower risk of developing stage II or II NEC. Could you explain your results and your discussion?

3) If, as the severity of NEC increases, the impact of HDP on NEC becomes more pronouced, why do you think HDP/preeclamspia/eclampsia didn't protect the VPIs against SIP?

6. PLOS authors have the option to publish the peer review history of their article (what does this mean?). If published, this will include your full peer review and any attached files.

Reviewer #1: No

Reviewer #2: **Yes: **Mariana Drechmer Romanowski

---

## [Author Response · Author response to Decision Letter 0]

15 Aug 2024

Dear editors and reviewers,

Thank you for your email and the reviewers' comments regarding our manuscript "Association of hypertensive disorder of pregnancy with necrotizing enterocolitis in very preterm infants: A retrospective cohort study" (PONE-D-24-12553). We have carefully considered the feedback and made the necessary revisions to our manuscript. Below, we address each point raised by the reviewers and the editor.

Reviewer #1 Comments:

1. Add the description about the missing data of NEC (L177-178):

Response: We have added a description about the missing data on NEC cases.

original sentence” After excluding 255 cases with multiple anomalies and 349 cases with missing maternal HDP data, 27,660 infants were analyzed.”

revised sentence” As there were no cases of NEC missing, after excluding 255 cases with multiple anomalies and 349 cases with missing maternal HDP data, 27,660 infants were analyzed.” (L177-178)

Reviewer #2 Comments:

1.Change the Word "Conference" (L172):

Response: We have corrected the spelling error in 'confidence.' Thank you for pointing it out.

2. Clarify the discrepancy between results and discussion regarding the correlation between maternal HDP and the occurrence of NEC stages II or III.

Response: We have clarified the discussion to align with the results, indicating no significant correlation between maternal HDP and NEC stages II or III after adjustments.

original sentence” Our study revealed that VPIs born to mothers with HDP had a lower risk of developing stage II or III NEC.”

revised sentence” Our study found no significant association between maternal HDP and the occurrence of stages II or III NEC in VPIs.” (L245-248)

3. If, as the severity of NEC increases, the impact of HDP on NEC becomes more pronounced, why do you think HDP/preeclampsia/eclampsia didn't protect the VPIs against SIP?

Response: We find exploring the relationship between HDP and SIP to be an intriguing topic worthy of further investigation. The evidence from our study is insufficient to support a protective role of HDP against SIP. Given that SIP and NEC involve different pathophysiological mechanisms and factors influencing their occurrence, exploring the association of HDP with SIP would require a new dedicated correlational study for verification.

Journal Requirements:

1. Manuscript Formatting: 

Response: We have ensured that our manuscript follows the PLOS ONE style templates.

2. ORCID ID:

Response: We have validated the corresponding author's ORCID ID in the Editorial Manager. 

3.Data Availability:

Response: The data underlying our findings are sourced from the Chinese Neonatal Network (CHNN) that we do not have the legal right to distribute. However, we provide some details to assist other researchers in accessing these data. 

4.Reference List:

Response: We have reviewed our reference list to ensure completeness and correctness. Our references do not include any retracted papers.

We believe these revisions have strengthened our manuscript, and we hope it now meets the criteria for publication in PLOS ONE. We look forward to your positive response.

Kind regards,

Yu Hu

---

## [Decision Letter · Decision Letter 1]

17 Oct 2024

Association of hypertensive disorder of pregnancy with necrotizing enterocolitis in very preterm infants: A retrospective cohort study

PONE-D-24-12553R1

Dear Dr. Hu,

We’re pleased to inform you that your manuscript has been judged scientifically suitable for publication and will be formally accepted for publication once it meets all outstanding technical requirements.

Kind regards,

Renato Teixeira Souza

Academic Editor

PLOS ONE

Additional Editor Comments (optional):

The authors have successfully addressed the points raised by the reviewers, and, in my opinion, it is ready for publication.

---

## [Editor Report · Acceptance letter]

22 Oct 2024

PONE-D-24-12553R1 

PLOS ONE

Dear Dr. Hu, 

I'm pleased to inform you that your manuscript has been deemed suitable for publication in PLOS ONE. Congratulations! Your manuscript is now being handed over to our production team.

Kind regards, 

on behalf of

Dr. Renato Teixeira Souza 

Academic Editor

PLOS ONE